# A Pathophysiological Intersection of Diabetes and Alzheimer’s Disease

**DOI:** 10.3390/ijms231911562

**Published:** 2022-09-30

**Authors:** Maša Čater, Sabine M. Hölter

**Affiliations:** 1Chair of Genetics, Animal Biotechnology and Immunology, Department of Animal Science, Biotechnical Faculty, University of Ljubljana, 1230 Domžale, Slovenia; 2Institute of Developmental Genetics, Helmholtz Munich, 85764 Neuherberg, Germany; 3School of Life Sciences, Technical University Munich, 85354 Freising, Germany

**Keywords:** diabetes, insulin sensitivity, Alzheimer’s disease, brain, cognition, depression

## Abstract

Diabetes is among the most prevalent diseases of the modern world and is strongly linked to an increased risk of numerous neurodegenerative disorders, although the exact pathophysiological mechanisms are not clear yet. Insulin resistance is a serious pathological condition, connecting type 2 diabetes, metabolic syndrome, and obesity. Recently, insulin resistance has been proven to be connected also to cognitive decline and dementias, including the most prevalent form, Alzheimer’s disease. The relationship between diabetes and Alzheimer’s disease regarding pathophysiology is so significant that it has been proposed that some presentations of the condition could be termed type 3 diabetes.

## 1. Introduction

The incidence of age-related metabolic and neurodegenerative diseases is increasing as the world population is aging [1,2]. Many studies confirmed that patients with impaired glucose tolerance and diabetes have an increased risk of developing Alzheimer’s disease (AD) compared to healthy individuals [3,4,5]. The association between diabetes and its effects on the development of retinopathy, neuropathy, nephropathy, and cardiovascular diseases is well established [6]. However, the underlying processes of how diabetes impacts the brain, cognition, and mental health are not yet fully understood. Therefore, in this review, we gathered up-to-date literature about the effects of impaired glucose metabolism on the brain, cognition, and mental health, and about the intersection of symptoms of diabetes and AD.

AD, the most common cause of dementia, is a progressive brain disorder that causes memory deficits and destroys cognitive functioning. A genetic mutation is proposed to be the cause for early-onset AD, while a sum of genetic, environmental and lifestyle factors are considered to trigger the late-onset AD [7]. Changes in the brain, that occur in the early stages of the disorder, include an abnormal buildup of proteins that form amyloid plaques and tau tangles. This results in neuron malfunctioning and a loss of connections between neurons, and eventually, in neuron death [8]. The damage in the brain is initiated in the hippocampus and the entorhinal cortex, both essential in memory formation. Therefore, the first signs of AD include memory problems and mild cognitive impairment. When damage spreads to other parts of the brain, it causes shrinkage of the brain. Thus, in the mild stage of AD, greater memory loss occurs with additional personality and behavior changes. Moreover, in moderate AD, language, reasoning, and sensory processing are affected by the damage in the brain. Hallucinations, delusions, and paranoia can occur at this stage as well. When AD progresses to a severe stage, with plaques and tangles that are wide-spread throughout the brain, communication and independent self-care become greatly affected [9].

AD and diabetes share many different pathologies, such as hyperglycemia, glucose intolerance, hyperinsulinemia, insulin resistance, adiposity, hypertension, atherosclerosis, and cognitive impairment [10,11,12]. The main bases for their development are proposed to be the malfunctioning of insulin signaling [13] with glucose metabolism disorders, deficits in mitochondrial activity, and cholesterol-associated pathologies—these are some of the many additional causes that have been observed by diabetes researchers and neuroscientists in recent decades [6,14,15,16,17]. In the following sections, we will look at each of the proposed factors causing diabetes and/or AD and find their correlations. Moreover, we will discuss the most frequently used animal models in diabetes and AD studies, which enabled the understanding of the mechanism of action of compromised insulin activity-related pathologies. This review is based on a PubMed search using combinations of the terms, «diabetes», «Alzheimer’s disease», «animal model», «mouse», «rat», «insulin» and «brain». Up-to-date references were selected by the date of publication (last 10–15 years, with exceptions) and by their main discovery conclusions so they fitted to the scope of this article in at least one of the aspects (diabetes or AD).

## 2. The Importance of Brain Insulin

The brain is the most energy-demanding organ that relies on glucose for fuel. Most of its energy expenditure is needed for maintaining the potential difference across the membranes of nerve cells—for dendritic and axonal transport, and tissue repair. The brain uses several pathways for gathering glucose. Glucose enters the brain by insulin-insensitive facilitated diffusion across the blood–brain barrier (BBB), followed by entering the brain cells using insulin-independent glucose transporters [18,19] as well as insulin-regulated glucose transporters [20].

Thus, insulin, a highly active hormone and a crucial polypeptide in the body, plays an important role in the brain (Figure 1). Like glucose, insulin also crosses the BBB, and when in the brain, insulin binds to insulin receptors on neurons and glial cells; there, its main function is to modulate glucose transfer into different brain cells for maintaining the normal functioning of the brain [21]. Additionally, insulin in the brain contributes to the control of nutrient homeostasis, reproduction, cognition, and memory, as well as to neurotrophic, neuromodulatory [22], and neuroprotective effects.

A third option for brain cells to gather glucose has been discovered in recent decades [23]. A fascinating detection of insulin mRNA transcripts in the brain revealed that the brain is capable of synthesizing insulin on its own [22,24,25].

### 2.1. Insulin Receptors

Insulin receptors are transmembrane receptors, activated by insulin and insulin-like growth factor I and II (IGF 1 and 2). Insulin receptors are expressed on all cell types—and in the brain—and are important for their role in inducing glucose uptake. The level of expression of insulin receptors varies between brain regions, being the highest in the olfactory bulb, hypothalamus, hippocampus, cerebral cortex, striatum, and cerebellum [21]. The main area associated with brain insulin is the hypothalamus, which is known to regulate feeding and energy balance [26]. Normal insulin signaling in the hypothalamus is important for maintaining metabolic homeostasis in peripheral tissues, such as liver and adipose tissue; the metabolic effects of brain insulin include the inhibition of glucose production in the liver, inhibition of lipolysis in adipose tissue, catabolism of branched-chain amino acids, and secretion of triglycerides in the liver [27,28]. Impaired metabolic control due to hypothalamic insulin dysfunction is, therefore, proposed to be one of the links between diabetes and brain dysfunction in AD [29].

### 2.2. Glucose Transporters

The brain is a large glucose consumer, hence, it uses both insulin-independent and insulin-dependent pathways for gathering glucose via different glucose transporters (GLUTs): GLUT1, GLUT3, and GLUT4. Glucose from the blood flow is transported across the BBB by GLUT1, which plays an important role in most body cells and in basal glucose uptake control, and is known to be insulin-insensitive. Both endothelial cells and astrocytes, which constitute the BBB, express GLUT1 as their main facilitative transporter for glucose uptake [18]. Astrocytes are a type of glial cell involved in neuroinflammation and are regulated by insulin regarding cytokine secretion in the presence of inflammatory stimuli [30]. Insulin modulates brain glucose metabolism by acting on astrocytes in cooperation with IGF 1. A combined action of insulin and IGF 1 has been shown to trigger a translocation of GLUT1 to the cell membrane of astrocytes [31]. The main purpose of this action is the recovery of neuronal activity after hypoglycemia. Therefore, the transport of glucose across the BBB using GLUT1 can be affected by obesity, glycemia, diabetes, inflammation, and levels of circulating triglycerides [32].

Furthermore, GLUT3 represents a major insulin-independent glucose transporter in neurons, which controls the basal glucose uptake in the brain [19]. Neural GLUT3-mediated glucose transport is stimulated not by insulin but by *N*-methyl-d-aspartate (NMDA) receptor-mediated depolarization. GLUT3 has a higher affinity for glucose and a much greater transport capacity than GLUT1 and GLUT4, which is of high importance as micro-environmental glucose levels at neurons are much lower than in serum [33].

GLUT4 is another glucose transporter involved with brain glucose metabolism. It is predominantly found in muscle cells and adipocytes. However, a co-expression of GLUT4 with GLUT3 has been determined in certain brain regions that are linked to cognitive behavior (basal forebrain, amygdala, hippocampus) and metabolic control (hypothalamus) [34]. GLUT4 is also important for glucose sensing and tolerance, and insulin sensitivity in general, due to its high presence in the hypothalamus [20]. In contrast to GLUT1 and GLUT3, GLUT4 is regulated by insulin and its major role in the brain is to accelerate glucose flux into neurons during periods of high metabolic demand [35]. Impairment in GLUT4 activity in the hippocampus results in reduced metabolism and plasticity of hippocampal neurons, provoking the development of depressive-like behavior and cognitive dysfunction [36,37], which can be observed in AD patients.

### 2.3. Brain Insulin Synthesis De Novo

Interestingly, several studies on laboratory animals [25,38,39] and humans [25,40] have suggested that insulin is synthesized de novo in certain brain regions and is important for maintaining the basal function of local circuits. Insulin mRNA transcripts have been detected in the hippocampus, hypothalamus, thalamus, and striatum. Hence, impaired synthesis of insulin in the brain is proposed to be one of many possible causes that trigger AD symptoms, as reduced levels of insulin mRNA transcripts in the hippocampus and hypothalamus have been shown in the brains of AD patients in post-mortem studies [23].

### 2.4. Insulin Effects on Specific Brain Cells

Apart from glucose uptake, insulin has many other roles in the brain. It affects different brain cells such as neurons, microglia, and astrocytes, as well as arterioles, capillaries, and BBB. A high density of insulin receptors (GLUT3) is located in neuronal synapses, whose functioning is the basis for cognition. Hence, the malfunctioning of GLUT3 in neurons and chronic hyperglycemia have been determined to negatively affect cognitive functioning, learning, and memory formation [41]. Insulin effects in neurons are very versatile, as insulin modulates catecholamine uptake and release, regulates expression and localization of ion channels—including GABA, NMDA, and α-amino-3-hydroxy-5-methyl-4-isoxazole propionic acid (AMPA) receptors—and regulates the activity of certain GLUT transporters. Insulin also modulates long-term potentiation and long-term depression via NMDA receptor signaling and AKT. Positive effects on neurite outgrowth, development of excitatory synapses, promoted dendritic spine formation, and improved neuronal survival are also important roles of insulin in the brain [21]. Moreover, the effects of insulin on oligodendrocytes are still not fully understood, yet certain studies propose that insulin might contribute to proliferation, differentiation, survival, and myelination of oligodendrocytes [42]. In contrast to neuronal insulin receptors, glial insulin receptors are downregulated in case of chronic hyperinsulinemia [43]. Astrocytes, which represent up to 40% of all glia cells in the human brain, bind insulin with a high affinity and take up glucose via GLUT1 [44]. Using glycolysis, astrocytes are capable of turning glucose into lactate and transport it to neurons as an additional energy source in case of high energy demand or hypoglycemia [18,45,46].

## 3. Involvement of Diabetes in the Development of Alzheimer’s Disease

### 3.1. Abnormal Protein Processing

The main feature of AD is abnormal protein processing in the brain, through which amyloid-β plaques and neurofibrillary tangles are formed, causing morphological pathologies in the brain tissue due to disintegrated microtubules, synaptic impairment, and neuronal apoptosis, thus, resulting in cognitive impairments and several psychopathologies (Figure 2).

Diabetes has been shown to be involved in the development of AD by affecting abnormal protein processing. Disturbed insulin signaling, associated with type 2 diabetes, affects the expression and metabolism of amyloid-β [47]. Moreover, abnormally phosphorylated tau protein, which induces the instability of neuronal microtubules and apoptosis of neurons in AD [48,49], has been observed in animal models of diabetes. Increased tau phosphorylation has been observed in type 1 and type 2 diabetes of mouse and rat models [50,51,52,53,54], as well as in humans with type 2 diabetes [55].

### 3.2. Deficient Insulin Signaling

Dysfunctional insulin receptor signaling is known to affect the expression and metabolism of amyloid-β and tau protein [47] and their clearance [56]. Insulin receptors are proposed to regulate synaptic activity as well; therefore, the malfunctioning of the receptors could cause neurodegeneration [57]. Moreover, insulin resistance in connection with hyperinsulinemia induces the accumulation of amyloid-β due to the lack of available insulin-degrading enzymes (IDEs). Normally, insulin and amyloid-β are both degraded by IDE. When insulin levels are increased, such as in type 2 diabetes, insulin uses the majority of IDE, and undegraded amyloid-β starts to accumulate in neurons [58]. As a possible treatment, insulin sensitizers have been tested in rodents [59] and early AD patients [60], with positive results in improving cognitive performances. Further studies are still needed to confirm the exact potential of insulin sensitizers for AD treatment. Apart from increased amyloid-β accumulation in neurons, hyperinsulinemia causes tau hyperphosphorylation in primary cortical neurons and hippocampal neurons [61,62,63], provoking their degeneration.

Alterations in insulin receptor signaling in type 2 diabetes and AD develop due to changes in both major signaling pathways. The mitogen-activated protein kinase (MAPK) pathway, required for cell proliferation, differentiation, and apoptosis [64], is accelerated in the brain of patients with AD [65]. The expression of MAPK co-localizes with aggregated tau in the hippocampus and cortical regions in AD brains, indicating that MAPK signaling is also involved in tau phosphorylation, synaptic plasticity, and neuroinflammation [66,67]. With respect to diabetes, Dusp8, which codes for a dual-specificity phosphatase involved in MAPK signaling and is predominantly expressed in the brain, was implicated by genome-wide association studies as a type 2 diabetes risk gene. There is evidence that Dusp8 can have sex-specific effects in mice and men on hypothalamic insulin resistance, hippocampal size, and cognitive, emotional, and hedonic behaviors [68,69,70]. The second insulin receptor signaling pathway, the Akt pathway—which is responsible for cell growth and survival, protein synthesis, and inhibition of the glycogen synthase kinase-3β (GSK-3β) enzyme [71,72,73,74,75]—is also affected by both AD and type 2 diabetes [76]. GSK-3β in the hippocampus and cortex is important for glycogenesis and glucose clearance. In normal conditions, its activity is inhibited by phosphorylation by insulin signaling via an insulin receptor. However, in type 2 diabetes, elevated activity of GSK-3β is proposed to trigger the reduction in glucose clearance by developing insulin resistance [75]. Moreover, increased GSK-3β activity is thought to result in increased amyloid-β production and tau phosphorylation [74,77]. Experiments in AD animal models and cell cultures have shown that GSK-3β is a good target for treatment development, as inhibiting GSK-3β successfully slowed down neurodegeneration [77,78].

### 3.3. The Cholinergic Hypothesis

Acetylcholine is a neurotransmitter that is involved in cholinergic neurotransmission. It is used by cholinergic neurons and has an important role in the peripheral and central nervous system, as cholinergic neurons are critical for cognition and memory. AD patients present with a continuous decline of cholinergic neurotransmission in their brain [79]. This occurs due to a decrease in the production of acetylcholine as well as the hydrolysis of acetylcholine. A mechanism of action has been proposed with a cholinergic hypothesis, which suggests that insulin plays an important role in acetylcholine production [80]. In case of hypoinsulinemia, less acetylcholine is produced due to a reduced expression of choline acetyltransferase (Figure 3). This direct effect of insulin on acetylcholine production has additionally strengthened the link between AD development, insulin malfunction, and diabetes; thus, AD has recently been considered a neuroendocrine disease and has been referred to as type 3 diabetes, possessing characteristics of type I and type II diabetes [80,81,82,83].

### 3.4. Glucose Metabolism Disorders

Apart from the impairments in brain insulin production and signaling, additional conditions are found at the intersection of diabetes and AD, such as oxidative stress and the formation of advanced glycation end products. Abnormal glucose metabolism and oxidative stress trigger the formation of advanced glycation end products which cause brain damage [14]. Advanced glycation end products are formed by the glycation of proteins or lipids in case of chronic hyperglycemia, and present a useful biomarker for degenerative diseases such as diabetes and AD. In normal aging, the formation of these molecules occurs in low levels, while it is greatly accelerated in patients with diabetes and AD [84,85]. Moreover, advanced glycation end products were found to induce the glycation of amyloid-β and tau, resulting in their formation and aggregation [86] (Figure 3). An increased expression of receptors for advanced glycation end products in neurons was determined in diabetic mice with impaired cognition [87], as well as at a clinical level in patients with AD and diabetes compared to non-diabetic AD patients [88].

### 3.5. Oxidative Stress

Impaired glucose metabolism has other effects in the body as well. It causes an accelerated production of free radicals, which results in oxidative stress in cells. Oxidative stress is well known to contribute to the development of diabetes and its neuropathies [89,90]. It has been indicated that oxidative cell damage occurs early in the development of AD [15], as increased concentrations of oxidized proteins in the hippocampus and in the frontal and parietal lobes were determined in patients with only a mild cognitive impairment. Oxidative stress is strongly linked to amyloid-β accumulation. Preclinical research in mice showed that antioxidant capacity decreases first, followed by an increase in lipid peroxidation, and finally, results in AD development [91,92].

Moreover, oxidative stress triggers local inflammations in the brain [93]. Some studies report that the use of anti-inflammatory drugs can decrease the risk of AD development, while others report no beneficial effects of taking these agents for treating AD [94,95,96].

### 3.6. Deficits in Mitochondrial Activity

Another factor correlating diabetes to AD is mitochondrial malfunctioning [6,16,76]. Mitochondrial activity is essential for normal neuronal functioning regarding ATP synthesis and for controlling calcium homeostasis [97]. While the efficiency of calcium homeostasis regulation decreases in the brain with normal aging, an increased calcium uptake by mitochondria has been observed among AD pathologies [98,99], as well as a decrease in mitochondrial mass and an increase in mitochondrial DNA in the cytoplasm [100]. Excessive calcium uptake triggers an increase in the level of reactive oxygen species and inhibition of ATP synthesis, resulting in neuronal degeneration and apoptosis. The exact mechanism of action regarding how mitochondrial malfunctioning is correlated to AD is not yet fully understood. However, it is proposed that mitochondrial electron transport is negatively affected by amyloid-β [101], which further on leads to mitochondrial malfunction and dysregulation of calcium homeostasis. Increased levels of intracellular calcium have been found to co-localize with neurofibrillary tangles and amyloid-β aggregates [99,102,103].

Interestingly, as excessive calcium uptake by mitochondria causes damage in neurons in AD, it initiates similar damage also in pancreatic cells, causing insulin malfunctions that lead to diabetes pathologies [104]. High levels of calcium in pancreatic β-cells are thought to trigger the malfunctioning of insulin secretion [105]. Insulin deficiency and oxidative stress due to a lower antioxidant capacity of neuronal mitochondria have been observed in type 1 diabetic rats [106], and these defects in mitochondrial DNA are determined to be inheritable. Moreover, in type 2 diabetes, a similar low antioxidant activity of mitochondria has been observed. However, in this type of diabetes, obesity is usually present, which is also known to be associated with smaller mitochondria and reduced energetic capacity [107].

### 3.7. Cholesterol-Associated Pathologies

The malfunctioning of cholesterol transportation within the circulatory system has been observed in diabetes and AD, yet the details of the underlying processes are unclear. In the case of diabetes, cholesterol has been found to be accumulated within pancreatic β-cells, causing a decrease in insulin secretion [108]. In AD mouse models, cholesterol has been determined at the same locations as amyloid-β plaques and tau proteins [109]. Therefore, it has been proposed that cholesterol is directly involved in the formation of protein abnormalities in AD (Figure 3).

Normal blood lipid levels are maintained by apolipoprotein E, which is expressed mainly in the brain and liver. Some alleles (*APOE* ε4) of the gene for apolipoprotein E result in the development of hypercholesterolemia and have been found in 40% of AD patients [76]. Additionally, the risk for AD development associated with *APOE* ε4 is doubled by diabetes [110]. Apolipoprotein E, synthesized from *APOE* ε4, is linked to abnormal protein processing, which is present in AD patients; in addition, apolipoprotein E is able to cooperate with amyloid-β aggregates and it promotes the phosphorylation of tau in neurons, inducing neurodegeneration [17,111].

## 4. Insulin Effects on Cognition and Mental Health

### 4.1. Cognitive Changes

Cognitive changes in people with dementia are well documented and studied. Recently, diabetes has received significant attention in psychiatry due to its high mutual frequency with mood disorders and cognitive impairment. Several symptoms of AD, such as cognitive dysfunctions regarding attention, memory, vocabulary, information processing, motor strength and speed, visual-motor and spatial skills as well as impaired general intelligence, were observed in patients with type 1 and type 2 diabetes [76,112]. Deficits in spatial learning and long-term potentiation in the hippocampus, which are important for memory formation, have been observed in patients with type 1 diabetes. Moreover, the most prevalent form of diabetes, type 2 diabetes, has been determined to trigger early cognitive and mental changes in a similar way as type 1 diabetes. While in type 1 diabetes the major problem causing physiological and cognitive deficits is insulin deficiency, in type 2 diabetes it is the malfunctioning of insulin receptor activity that has enormous consequences on the brain. Hence, insulin resistance, hyperinsulinemia, and impaired insulin signaling have been determined to cause many cognitive pathologies (Figure 4) [71,76].

The age of onset of diabetes, its duration, and degree of glycemic control are the main factors affecting how severe cognitive dysfunction will occur [113,114]. The degree of cognitive impairment progresses with long-lasting diabetes and poorly maintained glycemic control, and with the presence of diabetic complications such as depression and hypertension [21].

### 4.2. Mental Changes

Alzheimer’s disease is accompanied by depression and anxiety, which are the most prevalent mood disorders [115]. Diabetes often co-occurs with depression and anxiety, suggesting that insulin malfunctioning can play a role in mental changes. In association with the various effects on different brain cells, insulin has been shown to affect not only cognition, but also mood and psychiatric functioning [21]. The importance of brain insulin resistance is clearly seen in neuronal insulin receptor knock-out mice, which display spontaneous and life-long depressive-like and anxiety-like phenotypes [116].

The idea of influencing emotional behaviors with insulin dates back a century, when psychiatrists used insulin to cure mental illnesses by inducing a coma or as a shock treatment [117]. Now, we know that insulin receptors are highly present in the limbic region of the brain, where reward-based functioning, motivation, and emotions derive from. Many studies have proposed that diabetes and the development of mood disorders and their increased severity are causally linked [118]. One of the possibilities for the mechanism of action is the insulin modulation of brain serotonergic neurons and their neurotransmission, resulting in the development of anxiety and depressive symptoms [119]. Therefore, it is largely accepted that the risk of developing mood disorders is significantly increased in diabetic patients. Interestingly, there is a bidirectional correlation, as a 60% increase in risk for developing type 2 diabetes has been observed in depressed patients [120], yet a detailed mechanism of this correlation is not fully understood. Anhedonia is one of the domains of depression, defined as an incapacity to feel pleasure. It is suggested that anhedonia is associated with poor glycemic control in patients with type 2 diabetes. However, in patients with type 1 diabetes, a negative correlation between anhedonia symptoms and glycemic control was reported [121,122]. Further research is needed to clarify the mechanism of action which connects anhedonia with diabetes and AD.

## 5. Combined Effects of Antidepressants and Antidiabetic Drugs on Diabetes, Cognitive Impairment, and Mood Disorders

Interestingly, drugs for treating depression and diabetes were found to have a potential for treating conditions other than that of their primary purpose. Their potential often has joint properties, as some antidepressants help treating diabetes, while some antidiabetics are useful for treating cognitive impairment and mood disorders, which can develop in AD.

Preclinical and clinical studies have shown that serotonergic antidepressants help to improve diabetes and diabetic neuropathy. Antidepressants that increase serotonergic impact—for example, fluoxetine—promote the response to insulin [123,124]. However, no study has yet proposed a mechanism of action for such positive effects of selective serotonin reuptake inhibitors on glucose homeostasis. In contrast, tricyclic antidepressants and similar ones that impact on norepinephrine are known to increase insulin resistance and cause important cardiovascular side effects [125,126,127]. Therefore, it is of great importance to consider which class of antidepressant drugs to use due to their different impact on glucose homeostasis.

Anti-hyperglycemic agents were proposed to have great potential for improving cognition in diabetic patients and in patients with mood disorders [128]. Research in rats showed early stage memory formation deficits caused by diabetes can be successfully prevented by insulin treatment (41). Moreover, subcutaneous or intranasal insulin, glucagon-like peptide-1 agonists, and metformin present several antidepressant-like and antianxiety-like effects, bringing new targets for therapeutic options [119,129,130,131]. It remains to be seen if tirzepatide, a compound combining the activation of GLP-1 and glucose-dependent insulinotropic polypeptide receptors, also exerts a positive effect on mental health. Tirzepatide was recently approved by the FDA for the treatment of type 2 diabetes and reduces body weight in non-diabetic obese patients as well [132].

Combined use of antidepressants and antidiabetics can also be helpful in attenuating mood disorders in diabetic patients. Interestingly, antidiabetic drugs have been shown to positively affect depression-like behavior even in animals/people without metabolic impairments [133]. Several animal studies confirmed that drugs such as insulin, glyburide, metformin, pioglitazone, vildagliptin, liraglutide, and exenatide have both antidiabetic and antidepressant activities [134]. Clinical studies evaluating the antidepressant activity of antidiabetics in humans are limited and future clinical trials are required. Hence, antidiabetics show a promising new repurposed effect on the brain for treating neuropsychiatric disorders [135], which can develop in AD and diabetic patients by reducing blood glucose levels, ameliorating central oxidative stress and inflammation, modulating the hypothalamic–pituitary–adrenal axis, and enhancing serotonergic neurotransmission [119].

## 6. Animal Models for Diabetes and Alzheimer’s Disease Research

Several preclinical animal models have enabled diabetes and AD in vivo studies, ranging from insects to invertebrates, vertebrates, rodents, and other mammals. Fruit fly *Drosophila melanogaster*, nematode *Caenorhabditis elegans*, and zebrafish *Danio rerio* are some of the frequently used non-mammalian models which are particularly useful due to their short lifespan, low maintenance cost, conserved biochemistry, and whole-genome RNA interference libraries for performing high-throughput analyses of potential gene candidates involved in pathogenesis [136]. To overcome the main disadvantage of non-mammalian models regarding their distinct physiology and anatomy compared to humans, rodent models are in use and represent the major group of animal models. They are cost-effective and have a relative short life span. In general, rodents have a human-like physiology in many ways, yet researchers need to be careful about species-specific distinctions in physiology and anatomy. Large animal models exert a greater resemblance with human physiology and pharmacokinetics than rodents and non-mammals; however, their life cycle is long and specialized facilities are required for their housing, making the use of pigs and dogs costly. The best translational relevance is achieved with the use of non-human primate models, as their metabolic physiology and anatomy are most similar to humans. Their main disadvantage is that they have a long life cycle and are expensive to maintain. Additionally, there are only a limited number of approved facilities existing due to several ethical issues [136].

As mice and rats are the most widely used animals for biomedical research, we will focus on these preclinical animal models for studying metabolic disorders and neurological pathologies. There are many rodent models available for studying type 1 or type 2 diabetes and Alzheimer’s disease. New, improved animal models are continuously designed to mimic the diseases in people, yet none of them are a perfect analogue for people [137,138]. Interestingly, several animal models that were originally designed to study diabetes, obesity, and metabolic syndrome have shown a great potential for studies of neurodegeneration and AD [139]. Normally, rodents do not develop diabetes or AD; therefore, an induction of the development of pathologies is needed. There are several ways that diabetes and AD animal models are designed. In this review, we present those that are most often used, are designed by transgenic modifications, or induced by a compound, diet, or lesion. Transgenic modifications are often used to generate rodent models that simulate these human diseases. Genes of main interest for modifications are those that are involved in amyloid-β (*APP*, *PSEN1*) and tau (*MAPT*) development, lipid metabolism (*APOE ε4)*, and genes involved in insulin desensitization (*Lep*) [136,140,141] (Figure 5). However, other approaches for reproducing human diabetes and AD pathology in animals exist as well. Aging-based models develop the pathologies based on either natural aging or accelerated senescence [142,143].

### 6.1. Leptin Deficiency Rodent Models

One of the first mutant mouse lines involved in diabetes research were ob/ob and db/db mice. Ob mice develop mild diabetes but severe obesity, while db mice become moderately obese and severely diabetic [136]. The presence and importance of leptin was later identified in the development of obesity and diabetes [144], and the *Ob* gene was renamed *Lep*. Leptin deficiency is now known to importantly contribute to the development of cognitive decline in diabetic animal models. Mutations in the leptin gene *Lep* or its receptor *Lepr* are used to induce unregulated feeding, obesity, and diabetes in mouse models [140]. The ob/ob mouse has a spontaneous mutation in *Lep* which disables the secretion of leptin, while the db/db mouse has a spontaneous mutation in *Lepr*, resulting in the malfunctioning of leptin signal reception. Moreover, similar rat models with spontaneous defects in leptin signal reception are in use, such as the Zucker rat, the Koletsky rat, the ZDF rat, the Otsuka Long-Evans Tokushima Fatty rat, and others [136].

Leptin is a polypeptide hormone secreted by adipocytes and binds to the leptin receptor, which is highly expressed in the hippocampus, a brain area severely affected in AD. It is involved in neuroendocrine regulation of food intake and is involved in synaptic function and plasticity; therefore, it affects cognition and behavior [144,145,146]. Disruption of leptin metabolism or signaling plays an active role in the development of AD [147,148,149], although the concise pathways have not been determined yet. Moreover, leptin has been determined as a cognitive enhancer with neuroprotective capacities. Leptin has beneficial effects on learning and memory [150,151] and its mechanism of action is linked to the regulation of amyloid-β levels [152]. Impairments in hippocampal synaptic plasticity have been observed in genetically obese rodents with dysfunctional leptin receptors. Recent studies have demonstrated the potential of leptin as an AD therapeutic [153,154]. Leptin was two orders of magnitude more potent at reducing tau phosphorylation in neuronal cells with AD compared to insulin [155,156]. Direct administration of leptin into the hippocampus had a beneficial effect, as it facilitated long-term potentiation on site [157], thus, it had a great potential for reverting or preventing cognitive deterioration in AD patients.

### 6.2. Monogenic and Polygenic Diabetic Rodent Models

The severity of the induced diabetes in monogenic rodent models—such as leptin-deficient *ob*/*ob* and leptin receptor-deficient *db*/*db* mouse models, KK-A^y^ mice, and Otzhka Long-Evans Tokushima Fatty (OLETF) rats—is strongly affected by the strain’s genetic background. However, as diabetes and obesity are polygenic diseases in humans, an approach towards designing polygenic animal models has been made in recent decades. The Nagoya Shibata Yasuda mouse, the New Zealand obese (NZO) mouse, the Tsumura-Suzuki obese diabetes mouse, NONcNZO10/LtJ, and TALLYHO/JngJ, are some of the polygenic diabetic mouse strains becoming popular in diabetes, obesity, and AD research [158,159], due to the link between type II diabetes and a high risk of late-onset AD. Moreover, several polygenic rat models are being used for studying diabetes and associated diseases, including Goto-Kakizaki (GK) rats, DIO-sensitive Sprague Dawley rats, DR Sprague Dawley rats, UCD-T2DM rats, and Sand rats, which are actually gerbils [136,160]. Because of their polygenic nature, they are often considered better models of human type 2 diabetes than monogenic mutant animals.

### 6.3. Transgenic Rodents Models of Alzheimer’s Disease

Transgenic technology enabled the reproduction of the cause of familiar AD by transfecting a mutant human amyloid precursor protein (APP). There are several mouse models with mutations of human APP (PDAPP mice, APP mice, Tg2576 mice, PSAPP mice) (Figure 5) or human tau isoform inserted into the genome under the promoter (Tg-tau mice) that are in use in AD research [161,162,163]. However, some limitations persist in most of these murine models, such as a lack of progressive neuronal loss in the hippocampus and specific neocortical regions, that are present in a human AD brain, and other differences in the biochemical composition of plaques [164]. Additionally, there is a huge lack of mouse models for sporadic, late-onset AD, which represents more than 95% of AD cases [138,162]. Thus, a full reproduction of human AD features in mouse models yet remains to be accomplished in the future.

### 6.4. Streptozocin-Induced Hyperglycemic and AD Rodent Models

In rodents, a diabetes-like condition is often induced with the neoplastic agent streptozotocin, which triggers elevated blood glucose by injuring pancreatic β-cells [165,166]. An intraperitoneal injection of streptozotocin in transgenic mice prone to tau pathology results in an induction of type 1 diabetes and increased levels of hyperphosphorylated tau and amyloid-β [167,168]. Streptozotocin-induced hyperglycemia in rats has been shown to have a direct effect on their learning abilities and working memory [166]. On the other hand, induction of type 2 diabetes in transgenic mice prone to AD pathology caused no amyloid-β level increase, although it caused the early onset of cognitive dysfunction [169]. Therefore, it has been proposed that the early stages of diabetes-associated cognitive decline develop due to cerebral amyloid angiopathy, and not because of amyloid-β accumulation [76].

Dysfunctional insulin and IGF 1 signal pathways can trigger neurodegeneration in the brain [170], thus, streptozocin-induced animal models are also used in AD studies. Streptozocin injected intraperitoneally or intracerebroventricularly has been found to induce the pathologies of sporadic AD, such as insulin deficiency, plaque deposition, tau hyperphosphorylation, and neuronal loss [171,172,173]. However, streptozocin-induced AD animal models cannot reproduce all the AD pathologies found in humans, as they were found to lack neurofibrillary tangles [138].

### 6.5. Acrolein-Induced AD Animal Models

Similar to streptozocin, acrolein has been found to negatively impact the brain. Added in drinking water or administered through gavage, acrolein is used to induce AD pathologies in animal models. Acrolein is a reactive aldehyde and plays an important role in AD, as it causes nerve terminal damage and impairs neurotransmitter release [138,174].

### 6.6. Diet-Induced Diabetic and AD Rodent Models

Diet-induced obesity rodent models are available for studying obesity, the development of insulin resistance, and its pathologies. Animals are given free access to a high-fat diet or a high-sugar diet and are monitored for the development of diabetes-associated pathologies. The susceptibility to develop diet-induced obesity and diabetes is strain-specific. In mice, inbred C57BL/6J are widely used as a model for diet-induced obesity [175,176], while in rats, the most often used strains are outbred, such as Sprague Dawly, Wistar, and Long-Evans [136,177].

High-cholesterol diet-induced AD animal models have been established as well, as high cholesterol levels have been determined in AD patients [178]. Abnormal cholesterol metabolism is highly associated with the formation of plaques and tau phosphorylation [179]; therefore, a high-cholesterol diet is used to trigger cholesterol metabolic disorders in order to study AD in laboratory animals. Like in other AD animal models, diet-induced AD models are not perfect in simulating human AD pathologies. One of the disadvantages observed in high-cholesterol diet-induced AD animal models is a lack of hyperphosphorylated tau occurrence [138].

AD pathologies can also be induced by metal ion dysfunction. Metal ions are involved in several biochemical reactions in the body and their dysfunction leads to pathologies, disease development, and even death [180]. They can build complexes with plaques and produce reactive oxygen species. Moreover, metal ion dyshomeostasis can modulate enzymes and interrupt important biological functions such as ATP formation [138]. Animal models with an aluminum-induced AD by AlCl_3_ have an increased tau phosphorylation and oxidative stress with learning and memory deficits [181].

### 6.7. Lesion-Induced Diabetic and AD Rodent Models

Another approach for designing an animal model for diabetes and AD research is lesion-induced pathology. The hypothalamus is known as a key brain area for controlling metabolism, eating, and obesity development. Lesions in the ventromedial hypothalamus lead to overeating, weight gain, and adiposity. Alternative methods to make lesions involve cutting ventromedial hypothalamus axonal connections, stimulating this region via implanted electrodes, or injecting neuronal blockers. Recently, targeted genetic disruptions of certain brain regions or specific cells in the brain replaced the lesioning procedure [136]. Further on, traumatic brain injury-induced AD models mimic sporadic AD pathology in the brain and are usually designed by a controlled cortical impact model, a fluid percussion injury model, or a blast injury model [138]. These animal models have increased tau phosphorylation, plaque formations, white matter degradation, and BBB dysfunction [182,183].

## 7. Conclusions

The knowledge about the pathophysiological intersection of two highly prevalent diseases in the modern world, diabetes and AD, has broadened greatly in recent decades. Our comprehensive up-to-date literature review has found several intersection points, which show how neurodegenerative diseases, such as AD, trigger similar symptoms to those found in diabetic patients. The joint symptoms include abnormal glucose metabolism, neurodegeneration, cognitive impairment, and the development of mood disorders. The most important factors for the development of these symptoms are insulin signaling or synthesis malfunctioning, impairment of glucose transporters, and malfunctioning of cholesterol transportation within the circulatory system. The brain regions in which these impairments play a major role in provoking AD as well as diabetes, are the hypothalamus and the hippocampus. Dysregulated insulin signaling in the hypothalamus results in impaired metabolic control of liver and adipose tissue, while disrupted glucose transporters in the hippocampus lead to reduced neuronal plasticity, impaired cognition, and the development of depression. Hyperinsulinemia—as a result of insulin signaling dysfunction, oxidative stress in cells, and hypercholesterolemia—accelerates the production and accumulation of abnormally folded proteins in the brain, such as amyloid-β and tau protein, which have been confirmed to play a major role in the pathophysiology of AD. The hypothalamus and hippocampus are the main regions also important for brain insulin synthesis. Impaired brain insulin synthesis and malfunctioning of neuronal glucose transporters have been shown to negatively affect synapse activity and cause brain hypoinsulinemia and hyperglycemia. These conditions negatively affect the cholinergic neurotransmission and provoke the reduction in cognition and memory associated with AD. In conclusion, diabetes and AD are linked with several physiological pathways and pathological symptoms, yet not all mechanisms of action are understood. Further animal research as well as clinical investigations are needed to reveal the details of the connection between these two diseases and to find new potential target molecules or pathways for developing novel therapeutics.

## Figures and Tables

**Figure 1 ijms-23-11562-f001:**
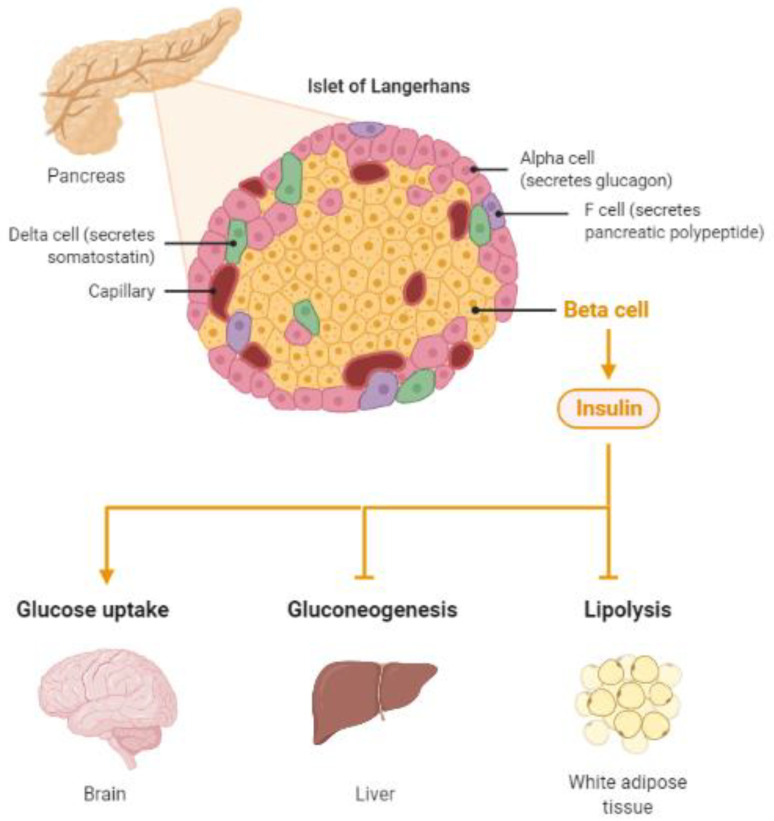
Insulin, a protein secreted by the pancreas, plays a major role in energy homeostasis of the body. Insulin pathologies are directly involved in the development of diabetes and Alzheimer’s disease (Created by BioRender.com, accessed on 23 August 2021).

**Figure 2 ijms-23-11562-f002:**
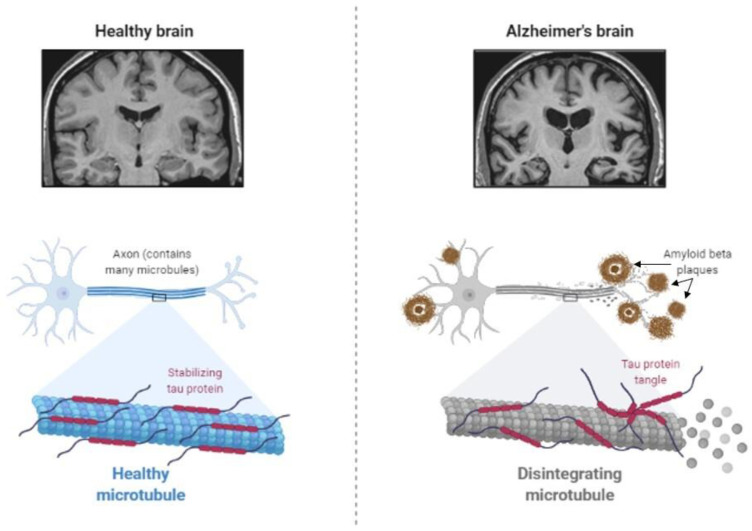
Abnormal protein processing occurs in the brain of patients with Alzheimer’s disease, which leads to the disintegration of microtubules and formation of tau aggregates and amyloid-β plaques, causing morphological pathologies, several cognitive impairments, and behavioral changes (Created by BioRender.com, accessed on 23 August 2021).

**Figure 3 ijms-23-11562-f003:**
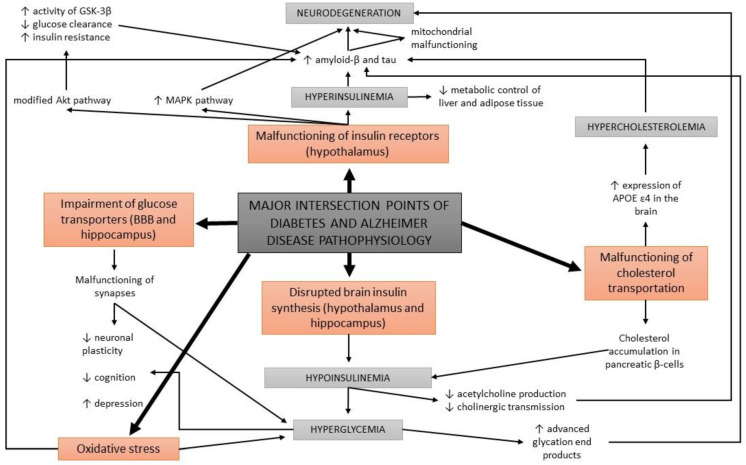
A schematic overview of the major intersection points of diabetes and Alzheimer’s disease pathophysiology.

**Figure 4 ijms-23-11562-f004:**
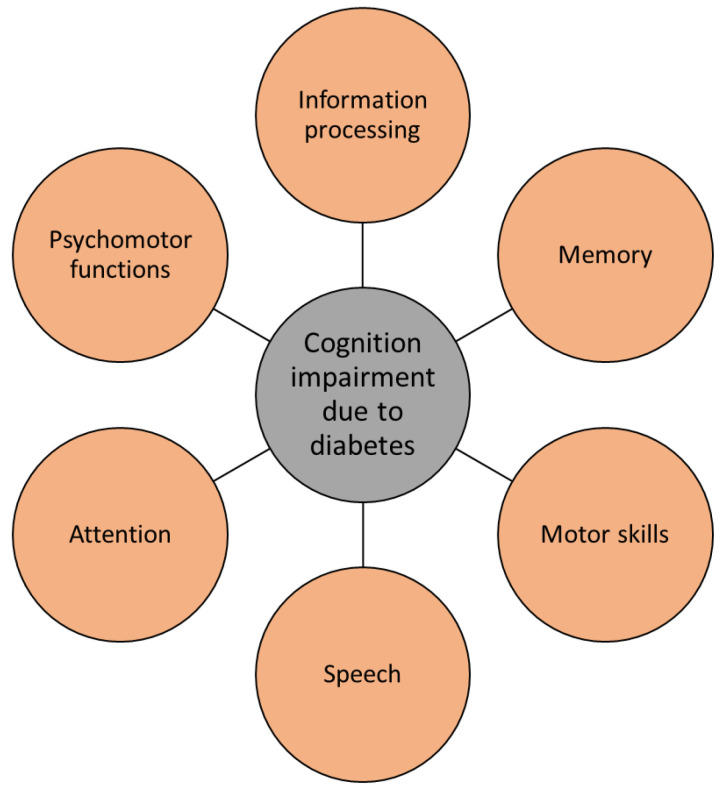
Diabetes causes several cognitive impairments, typical for Alzheimer’s disease.

**Figure 5 ijms-23-11562-f005:**
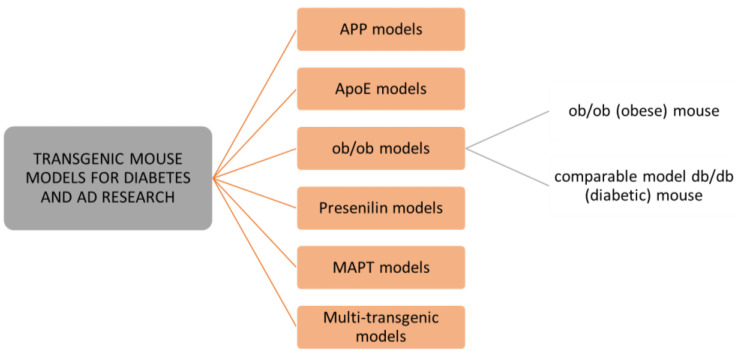
Overview of the most common transgenic mouse models used for studying diabetes and Alzheimer’s disease.

## Data Availability

Not applicable.

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
