# Peer review of "A Pathophysiological Intersection of Diabetes and Alzheimer’s Disease"

_ijms, 2022, doi:10.3390/ijms231911562_

Round 1

Reviewer 1 Report

This review is a comprehensive survey of major scientific databases for information on the pathophysiological intersection of diabetes and Alzheimer’s disease based on a very detailed review of relevant scientific literature. The work is very interesting and well-organized; I really enjoyed reading it. I only have some minor observations that are listed in the following lines.

1.       Methodology: Which databases were used? How were the references for this review selected? What were the reference inclusion and exclusion criteria?

2.       Line 37: I think it is better to use the word “sections” instead of “chapters.”

3.       Lines 52-54: provide a reference.

4.       Avoid using abbreviations in figure legends.

5. Add a list of abbreviations

Reviewer 2 Report

In this manuscript, the authors provided an overview of accumulating evidence of the association between diabetes and Alzheimer's disease (AD). 

Line 10. This review was supposed to focus on the association between diabetes and AD. Obesity does not fit well with the scope of this review.

Line 23. The word "correlation" should be changed to "association" because there was not statistical data to back up this statement.

Line 47 - 54. This paragraph was poor written. I has a difficult time trying to figure out what the authors were trying to say. "Glucose insertion" should be "glucose transfer". Was there a reference to support the study described in the last sentence of the paragraph (Line 52-54).

Line 111. Reference #19 did not have the data showing reduced insulin mRNA levels in post-mortem brain samples of AD patients. Please provide the correct reference. 

Line 202-203. This statement was not accurate. AD is referred to as type 3 diabetes NOT because of the effect of insulin on acetylcholine production. The two references cited did not support this statement either. 

Line 249-257. This paragraph was confusing. It didn't indicate if any of the mechanisms mentioned was involved in the contribution of insulin intolerance to AD pathogenesis. 

Section 6.1 (Line 406) and 6.2 (Line 435). Have those two models ever been used to study insulin intolerance in AD? If not, they should not be included in this review. 

Reviewer 3 Report

 The review explains the potential correlation between diabetes and Alzheimer’s disease, which also give a new prospect for the uncertain mechanism of this terrible brain disease. However, there are some problems:

 1.     The author would better add a new paragraph to describe the pathology, ethology and psychology of AD diseases briefly.

2.     In the introduction section, the last sentence is too long and hard to understand. Please just rephrase properly and explain the purpose explicitly.

3.     In the section called “The importance of brain insulin”, the author mentions that insulin is an active compound. However, from a biological perspective, the essence of insulin is actually a polypeptide.

4.     In addition, in the same section above, as the author write, “A fascinating detection insulin mRNA transcripts in the brain revealed that the brain is capable of synthesizing insulin on its own.”, are there any some research articles or review to support that there are indeed mRNA transcripts of insulin of the brain? Sometime, even mRNA transcripts are detected by RNA sequence technology, the corresponding proteins cannot be detected. This phenomenon does exist.

5.     Line 136: The name of section is not precise. This section describes the correlation between diabetes and the potential factors contributing to the occurrence of AD. “Protein folding” may raise misunderstanding.

6.     Line 137-147: The descriptions of the abnormal protein processing and their correlations with diabetes shouldn’t be put at the beginning of section 3. Since the whole section, as mentioned above, describes the correlation between diabetes and the potential factors, these words should be a new sub-section in section 3, rather than the introductory part.

7.     In the Figure 2, the location of amyloid-β plaques is not clear.

8.     Line 307-308: The description of AD disease is too long. Just write the key point that Alzheimer’s disease accompanies with depression and anxiety as the most prevalent ones.

9.     In section 6, the authors mainly introduce the animals related to obesity and diabetes. Though some of them have defects in learning and cognitive ability, the animal models designed for AD is not contained.

Round 2

Reviewer 2 Report

The authors have addressed all my concerns. 

Author Response

The English language has been corrected by one of our colleagues who is a native English speaker, all changes are tracked in the submitted revision.